# Differentiable Tree Search in Latent State Space

## Abstract

In decision-making problems with limited training data, policy functions approximated using deep neural networks often exhibit suboptimal performance. An alternative approach involves learning a world model from the limited data and determining actions through online search. However, the performance is adversely affected by compounding errors arising from inaccuracies in the learnt world model. While methods like TreeQN have attempted to address these inaccuracies by incorporating algorithmic structural biases into their architectures, the biases they introduce are often weak and insufficient for complex decision-making tasks. In this work, we introduce *Differentiable Tree Search* (DTS), a novel neural network architecture that significantly strengthens the inductive bias by embedding the algorithmic structure of a best-first online search algorithm. DTS employs a learnt world model to conduct a fully differentiable online search in latent state space. The world model is jointly optimised with the search algorithm, enabling the learning of a robust world model and mitigating the effect of model inaccuracies. We address potential Q-function discontinuities arising from naive incorporation of best-first search by adopting a stochastic tree expansion policy, formulating search tree expansion as a decision-making task, and introducing an effective variance reduction technique for the gradient computation. We evaluate DTS in an offline-RL setting with a limited training data scenario on Procgen games and grid navigation task, and demonstrate that DTS outperforms popular model-free and model-based baselines.

## 1 Introduction

Deep Reinforcement Learning (DRL) has advanced significantly in addressing complex sequential decision-making problems, largely due to advances in Deep Neural Networks (DNN) (Silver et al., 2017b; Berner et al., 2019; Vinyals et al., 2019). The superior representational capacity of DNNs enables an agent to learn a direct mapping from observations to actions through a policy (Mnih et al., 2016; Schulman et al., 2015b; 2017; Cobbe et al., 2021) or Q-value function (Mnih et al., 2013; Wang et al., 2016; van Hasselt et al., 2016; Hessel et al., 2018; Badia et al., 2020). However, DRL has high sample complexity and weak generalisation capability that limits its wider application in complex real-world settings, especially when a limited training data is available.

Model-based Reinforcement Learning (MBRL) approaches (Kaiser et al., 2020; Hafner et al., 2020; Ha & Schmidhuber, 2018) address this by learning world models from limited environment interactions and subsequently using these models for online search (Hafner et al., 2020; 2019) or policy learning (Kaiser et al., 2020). Although MBRL is effective for problems where learning the world model is simpler than learning the policy, its efficacy diminishes for complex, long-horizon problems due to accumulated errors arising from inevitable inaccuracies in the learnt world model.

Some recent attempts (Racanière et al., 2017; Silver et al., 2017a; Lee et al., 2018; Tamar et al., 2016; Farquhar et al., 2018; Guez et al., 2019) have tried to improve the sample efficiency and generalisation capabilities of DNN models by incorporating algorithmic structural biases into the network architecture. These inductive biases restrict the class of functions a model can learn by leveraging domain-specific knowledge, reducing the likelihood of the model overfitting to the training data and learning a model incompatible with the domain knowledge. Despite their potential, the weaker structural biases imposed by current methods limits the extent of their benefits.

A notable recent work, TreeQN (Farquhar et al., 2018), combines look-ahead tree search with deep neural networks. It dynamically constructs a computation graph by fully expanding the search tree up to a predefined depth $d$ using a learnt world model and computing the Q-values at the root node by recursively applying Bellman equation on the tree nodes. The whole structure is trained end-to-end and enables learning a robust world model that is helpful in the look-ahead tree search. Consequently, TreeQN outperforms conventional neural network architectures on multiple Atari games. However, the size of the full search tree grows exponentially in the depth, which computationally limits the TreeQN to perform only a shallow search. This issue limits TreeQN's ability to fully exploit the domain knowledge of advanced online search algorithms that can handle problems requiring much deeper search.

In this paper, we extend the foundational concepts of TreeQN, and propose Differentiable Tree Search (DTS), a novel neural network architecture that addresses these limitations by embedding the algorithmic structure of a best-first search algorithm into the network architecture. DTS has a modular design and consists several learnable submodules that dynamically combine to construct a computation graph by following a best-first search algorithm. As the computation graph is end-to-end differentiable, we jointly optimise the search submodules and the learnt world model using gradient-based optimisation. Joint optimisation enables learning a robust world model that is useful in the online search and optimises search submodules to account for the inaccuracies in the learnt world model.

However, a naive incorporation of a best-first search algorithm in the network design might lead to a Q-function which is discontinuous in the network's parameter space. This is because a small change in parameters might result in a different sampled tree and might bring large change in output Q-value. Consequently, the Q-function might be difficult to optimise as gradient-based optimisation techniques generally requires a continuous function for optimisation. To address this, we propose employing a stochastic tree expansion policy and optimising the expected Q-value at the root node, ensuring the continuity of Q-function in the network's parameter space. Furthermore, we propose formulating the search tree expansion as a decision-making task with the goal of progressively minimising the prediction error in the Q-value, and refining the tree expansion policy via REINFORCE (Williams, 1992). The REINFORCE algorithm has high variance in its gradient estimate and we further propose the use of the telescoping sum trick used in Guez et al. (2018) as an effective variance reduction method.

We compare DTS against popular baselines that include model-free Q-network, to assess the impact of incorporating algorithmic structural biases into the network architecture; TreeQN, to assess the benefits of incorporating a better online search algorithm capable of performing a deeper search within the same computational budget; and an online search with a learnt world model, to evaluate the advantages of jointly learning the model with the search algorithm. We evaluate these methods on deterministic decision making problems, including Procgen games (Cobbe et al., 2020) and a small-scale 2D grid navigation problem, to examine their generalisation capabilities. Empirical evaluations demonstrate that DTS outperforms the baselines in both Procgen and navigation tasks.

## 2 RELATED WORKS

In recent years, Reinforcement Learning (RL) has undergone remarkable advancements, notably due to the integration of Deep Neural Networks (DNNs) in domains such as Q-learning (Mnih et al., 2013; Wang et al., 2016; van Hasselt et al., 2016; Hessel et al., 2018; Espeholt et al., 2018; Badia et al., 2020) and policy learning (Williams, 1992; Schulman et al., 2015b; 2017; Mnih et al., 2016). Moreover, model-based RL has been significantly enriched by a series of innovative works (Deisenroth & Rasmussen, 2011; Nagabandi et al., 2018; Chua et al., 2018) that learn World Models (Ha & Schmidhuber, 2018) from pixels and subsequently utilise them for planning (Hafner et al., 2020; 2019; Schrittwieser et al., 2020) and policy learning (Kaiser et al., 2020).

An intriguing dimension of research in RL seeks to merge learning and planning paradigms. AlphaZero (Silver et al., 2017b) utilises DNNs as heuristics with Monte Carlo Tree Search (MCTS). On the other hand, use of inductive biases (Hessel et al., 2019) has been explored for their impact on policy learning. Recent works like Value Iteration Network (VIN) (Tamar et al., 2016) represents value iteration algorithm in gridworld domains using a convolutional neural network. Similarly, Gated Path Planning Network (Lee et al., 2018) replaces convolution blocks in VIN with LSTM blocks

to address vanishing or exploding gradient issue. Further, Neural A* Search (Yonetani et al., 2021) embeds A* algorithm into network architecture and learns a cost function from a gridworld map. Alternatively, Neural Admissible Relaxation (NEAR) (Shah et al., 2020) learns an approximately admissible heuristics for A* algorithm. Works like Predictron (Silver et al., 2017a) and Value Prediction Network (Oh et al., 2017) employs a learnt world model to simulate rollouts and accumulate internal rewards in an end-to-end framework. Extending this paradigm, Imagination-based Planner (IBP) (Pascanu et al., 2017) utilises an unstructured memory representation to collate information from the internal rollouts. Alternatively, MCTSnets (Guez et al., 2018) embeds the structure of Monte Carlo Tree Search (MCTS) in the network architecture and steers the search using parameterised memory embeddings stored in a tree structure. However, Guez et al. (2019) suggests that recurrent neural networks could display certain planning properties without requiring a specific algorithmic structure in network architecture.

Our work on DTS draws inspiration from these recent contributions, notably TreeQN (Farquhar et al., 2018). TreeQN employs a full tree expansion up to a fixed depth, followed by recursive updates to the value estimate of tree nodes in order to predict the Q-value at the input state. However, TreeQN's full tree expansion is exponential in the depth, rendering it computationally expensive for tackling complex planning problems that necessitate deeper search. DTS addresses this issue by incorporating a more advanced online search algorithm that emphasises expansion in promising areas of the search tree.

## 3 DIFFERENTIABLE TREE SEARCH

Differentiable Tree Search (DTS) is a neural network architecture that incorporates the structural bias of a best-first search algorithm into the network design. Moreover, DTS employs a learnt world model to eliminate the dependency on availability of a world simulator. The learnt world model is jointly optimised with the online search algorithm using gradient-based optimisation techniques, such that the learnt world model, although imperfect, is useful for the online search, and search submodules are robust against errors in the world model.

### 3.1 LEARNABLE SUBMODULES

DTS is built upon several learnable submodules, which are employed as subroutines in alignment with a best-first search algorithm to dynamically construct the computation graph. *Encoder Module* ($\mathcal{E}_\theta$) encodes an actual state $s_t$ into a latent state representation $h_t$, where $h_t = \mathcal{E}_\theta(s_t)$, and enables conduction of online search in a latent space. *Transition Module* ($\mathcal{T}_\theta$) approximates the transition function of the underlying model. It uses latent state $h_t$ and action $a_t$ as inputs to predict latent state $h_{t+1}$ of the next state, i.e. $h_{t+1} = \mathcal{T}_\theta(h_t, a_t)$. *Reward Module* ($\mathcal{R}_\theta$) approximates the reward function of the underlying model. Similar to the transition module, it takes latent state $h_t$ and action $a_t$ as inputs, and predicts the corresponding reward $r_t$ for the transition, i.e. $r_t = \mathcal{R}_\theta(h_t, a_t)$. *Value Module* ($\mathcal{V}_\theta$) approximates the value function of the underlying model by mapping latent state $h_t$ to its estimated state value $\mathcal{V}_\theta(h_t)$.

By breaking the network into these submodules and reusing them, the total number of learnable parameters is effectively reduced, helping to prevent overfitting to an arbitrary function that may align with the given training data.

### 3.2 ONLINE SEARCH IN LATENT SPACE

The search begins by encoding the input state $s_0$ into its latent state $h_0$ at the root node. Following this, Differentiable Tree Search proceeds in two stages. In the expansion phase, DTS iteratively expands the search tree from the input state for a fixed number of node expansions (each expansion is referred to as search trial). The expansion phase is followed by the backup phase, where the Q-values at the root node are recursively computed by applying the Bellman equation on the expanded tree nodes. Each node in the tree represents a latent state reachable from the root node, whereas each branch represents an action taken at the tree node. A set of candidate nodes, $O$, is maintained during the expansion phase, representing the tree nodes eligible for further expansion. The pseudo-code of DTS is presented in Algorithm 1 in the appendix.

In the **Expansion phase**, each trial begins by evaluating the total path value, $\bar{V}(N)$, of the candidate nodes. The total path value is the cumulative sum of rewards from the root node to a particular leaf node $N$, in addition to the value of the leaf node predicted by the value module, $\mathcal{V}_\theta(h_N)$, i.e.

$$\bar{V}(N) = \mathcal{R}_\theta(h_0, a_0) + ... + \mathcal{R}_\theta(h_{N-1}, a_{N-1}) + \mathcal{V}_\theta(h_N) \tag{1}$$

The node $N^*$ with the highest total path value is selected for expansion in the deterministic search version; for differentiable search, a node is sampled from the candidates using a distribution constructed with the softmax of the path values of the candidates. This expansion is carried out by simulating every action on the node $N^*$ using the Transition module, $\mathcal{T}_\theta$. Simultaneously, the associated reward, $\mathcal{R}_\theta(h_{N^*}, a)$, is derived. The resulting latent states are incorporated into the tree as children of the node $N^*$. Additionally, they are added to the candidate set $O$ to be considered for subsequent expansions, while $N^*$ is excluded from the set. This can be represented as:

$$O \leftarrow O \cup h_a | \; h_a = \mathcal{T}_\theta(h_{N^*}, a); \forall a \in A - N^*$$

The expansion phase is followed by the **Backup phase**. In this phase, value of all the tree nodes is recursively updated using the Bellman equation as follows:

$$Q(N, a) = \mathcal{R}_\theta(h_N, a) + V(\mathcal{T}_\theta(h_N, a))$$

$$V(N) = \begin{cases} \mathcal{V}_\theta(h_N), & \text{N is a leaf node} \\ \max_a Q(N, a), & \text{otherwise} \end{cases}$$

After the Backup phase, the Q-values at the root node are returned as the final output of the online search.

Throughout the Expansion and Backup phases, a dynamic computation graph is constructed where the output Q-values depend on the submodules, namely Encoder, Transition, Reward, and Value modules. During training, the output Q-values are evaluated by a loss function. Gradient-based optimisers such as Stochastic Gradient Descent (SGD) backpropagates the gradient of this loss through the entire computation graph, and updates the parameters of each submodule in an end-to-end process.

### 3.3 DISCONTINUITY OF Q-FUNCTION

For the effective optimisation of a neural network's parameters using gradient descent, it is desirable for the network's output to be continuous in the parameter space. We first show that a network which is constructed from a full tree expansion to a fixed depth $d$, as used in TreeQN, is continuous.

**Theorem 3.1.** *Given a set of parameterised submodules that are continuous within the parameter space, expanding a tree fully to a fixed depth 'd' by composing these modules and computing the Q-values by backpropagating the children values using addition and max operations, is continuous. (See Section B for the proof.)*

Composing this Q-function with a continuous loss function ensures its continuity, thereby facilitating gradient-based optimisation. However, DTS approximates the full tree expansion by only expanding the paths that are likely to represent the optimal trajectory from the root node. When the network parameters are changed slightly, DTS might generate a different tree structure, which, in turn, would impact the Q-value computed at the root node, and subsequently, the continuity of DTS.

### 3.4 STOCHASTIC TREE EXPANSION POLICY

To make the loss function continuous with respect to the network's parameters, we use the expected loss with respect to a stochastic tree expansion policy. Let us represent a partially sampled tree after $t$ trials as $\tau_t$. The output Q-values of DTS, $Q_\theta(s, a | \tau)$, depends on the final tree $\tau$ sampled after $T$ trials of the online search. We can construct a stochastic tree expansion policy $\pi_\theta(\tau_t)$ which takes a tree $\tau_t$ as input and outputs a distribution over the candidate nodes, facilitating selection of a node for further expansion and generating the tree $\tau_{t+1}$. We compute the stochastic tree expansion policy by taking softmax over the total path value (as defined in equation 1) of each candidate node $n_t \in O(\tau_t)$ for the tree $\tau_t$, i.e.

$$\pi_\theta(n_t | \tau_t) = \text{softmax}_{n_t} \bar{V}_\theta(n_t); \quad \text{where } n_t \in O(\tau_t) \tag{2}$$

Let us represent the loss function on the output Q-value as $\mathcal{L}\Big(Q_\theta(s, a|\tau)\Big)$. Given our aim to optimise the expected loss via gradient descent, the gradient of the expected loss (Schulman et al., 2015a) can be computed as follows (See Section C for the derivation):

$$\mathcal{L} = \mathbb{E}_\tau \left[ \mathcal{L}\Big(Q_\theta(s, a|\tau)\Big) \right] \tag{3}$$

$$\nabla_\theta \mathcal{L} = \nabla_\theta \mathbb{E}_\tau \left[ \mathcal{L}\Big(Q_\theta(s, a|\tau)\Big) \right] \tag{4}$$

$$= \mathbb{E}_\tau \left[ \mathcal{L}\Big(Q_\theta(s, a|\tau)\Big) \sum_{t=1}^{T} \nabla_\theta \log \pi_\theta(n_t|\tau_t) + \nabla_\theta \mathcal{L}\Big(Q_\theta(s, a|\tau)\Big) \right] \tag{5}$$

The expected loss, as depicted in equation 3, is continuous in the parameter space $\theta$ and can be optimised using well-known gradient-based optimisation techniques. For empirical evaluations, we use a single sample estimate of the expected gradient in equation 4.

### 3.5 REDUCING VARIANCE USING TELESCOPIC SUM

The REINFORCE term of the gradient in equation 5, i.e. $\mathcal{L}\Big(Q_\theta(s, a|\tau)\Big) \sum_{t=1}^{T} \nabla_\theta \log \pi_\theta(n_t|\tau_t)$, usually has high variance due to the difficulty of credit assignment (Guez et al., 2018) in a reinforcement learning type objective; the second part of the gradient equation is the usual optimisation of a loss function, so we expect it to be reasonably well behaved. To reduce the variance of the first part of the gradient, we take inspiration from the telescoping trick in Guez et al. (2018).

Let us denote the value of loss after $t^{th}$ search trial as $\mathcal{L}_t = \mathcal{L}\Big(Q_\theta(s, a|\tau_t)\Big)$. The objective is to minimise (or equivalently, maximise the negative of) the loss value after $T$ search trials, represented as $\mathcal{L}_T$. Assuming that $\mathcal{L}_0 = 0$, we can rewrite $\mathcal{L}_T$ as a telescoping sum:

$$\mathcal{L}_T = \mathcal{L}_T - \mathcal{L}_0 = \sum_{t=1}^{T} \mathcal{L}_t - \mathcal{L}_{t-1}$$

Now, let us define a reward term, $r_t$, for selecting node $n$ during the $t^{th}$ search trial as the reduction in the loss value post tree expansion in the $t^{th}$ trial, i.e. $r_t = \mathcal{L}_t - \mathcal{L}_{t-1}$. Further, let us represent the return or the sum of rewards from trial $t$ to the final trial $T$ as $R_t$, which can be computed as:

$$R_t = \sum_{i=t}^{T} r_i = \mathcal{L}_T - \mathcal{L}_{t-1}$$

Given this, the REINFORCE term from equation 5 can be reformulated as $\sum_{t}^{T} \nabla_\theta \log \pi_\theta(n_t|\tau_t) R_t$, where $\mathcal{L}_{t-1}$ serves as a baseline to help reduce variance. Consequently, the final gradient estimate of the loss in equation 3 is expressed as:

$$\nabla_\theta \mathcal{L} = \mathbb{E}_\tau \left[ \sum_{t}^{T} \nabla_\theta \log \pi_\theta(n_t|\tau_t) R_t + \nabla_\theta \mathcal{L}\Big(Q_\theta(s, a|\tau)\Big) \right] \tag{6}$$

### 3.6 AUXILIARY LOSS FUNCTIONS FOR WORLD MODEL CONSISTENCY

During the online search, the transition, reward, and value networks operate on the latent states. Consequently, it is essential to maintain a consistent scale for the inputs to these networks (Ye et al., 2021). To achieve this, we apply Tanh normalisation on the latent states, adjusting their scale to be within the range $(-1, 1)$. Additionally, as the search is performed in the latent space, we incorporate self-supervised consistency loss functions (Schwarzer et al., 2021; Ye et al., 2021) to ensure consistency in the transition and reward networks.

Consider actual states $s_t$ and $s_{t+1}$, where $s_{t+1}$ is obtained by taking action $a_t$ in state $s_t$. Their corresponding latent state representations denoted as $h_t$ and $h_{t+1}$. Here, $h_t = \mathcal{E}_\theta(s_t)$ and $h_{t+1} = \mathcal{E}_\theta(s_{t+1})$. Now, we can use the transition module to predict another latent representation of

state $s_{t+1}$, represented as $\bar{h}_{t+1} = \mathcal{T}_\theta(h_t, a_t)$. To ensure that the transition function, $\mathcal{T}_\theta$, provides consistent predictions for the transitions in the latent space, we minimise the squared error between the latent representations $h_{t+1}$ and $\bar{h}_{t+1}$.

$$\mathcal{L}_{\mathcal{T}_\theta} = \mathbb{E}_{(s_t,\, a_t,\, s_{t+1}) \sim \mathcal{D}} \Big( \bar{h}_{t+1} - h_{t+1} \Big)^2 \tag{7}$$

In a similar vein, we minimise the mean squared error between the predicted reward $\mathcal{R}_\theta(h_t, a_t)$ and the actual reward observed $r_t$ in the training dataset $\mathcal{D}$.

$$\mathcal{L}_{\mathcal{R}_\theta} = \mathbb{E}_{(s_t,\, a_t,\, r_t) \sim \mathcal{D}} \Big( \mathcal{R}_\theta(h_t, a_t) - r_t \Big)^2 \tag{8}$$

Additional details on these loss functions are presented in Section D in the appendix.

## 4 Experiments

### 4.1 Test Domains

**Navigation:** This is a 2D grid-based navigation task designed to quantitatively and qualitatively visualise the agent's generalisation capabilities. The environment is a $20 \times 20$ grid featuring a central hall. At the start of each episode, a robot is randomly positioned inside this hall, while its destination is set outside. We present two distinct scenarios: one with a single exit and another with two exits from the central hall. Training is done only in the two exit scenario and the single exit scenario, which requires a longer-horizon planning to reach the goal, is used to test generalisation.

**Procgen:** Procgen is a collection of 16 procedurally generated, game-like environments, specifically designed to evaluate an agent's generalisation capability, differentiating it from Atari 2600 games (Mnih et al., 2013). The open-source code for these environments can be found at `https://github.com/openai/procgen`. Further details on these domains are presented in Section E of the appendix.

### 4.2 Learning Framework

Differentiable Tree Search (DTS) can serve as a drop-in replacement for conventional Convolutional Neural Network (CNN) architectures. It can be trained using both online and offline reinforcement learning algorithms. In this paper, we employ the offline reinforcement learning (Offline-RL) framework to focus on the sample complexity and generalisation capabilities of DTS when compared with the baselines. Offline-RL, often referred to as batch-RL, is the scenario wherein an agent learns its policy solely from a fixed dataset of experiences, without further interactions with the environment. We outline the dataset compilation process and the loss function used for training in subsequent sections.

#### 4.2.1 Training Datasets

We used an expert/optimal policy to collect the offline training dataset. The training dataset, $\mathcal{D}$, consists of expert trajectories generated using the expert policy $\pi^*$, where each trajectory is defined as a series of $T$ tuples, each comprising the state observed, action taken, reward observed, and the target Q-value of the observed state, denoted as $(s_t, a_t, r_t, Q_t)$. The target Q-value for state $s_t$ can be computed by adding the rewards obtained in the trajectory from timestep $t$ onwards, i.e. $Q_t = Q^{\pi^*}(s_t) = \sum_{i=t}^{T} r_i$. For our experiments, we collected 1000 trajectories for each test domain.

#### 4.2.2 Loss Function

The primary objective is to have the Q-values, as computed by the agent, closely approximate the target Q-values for corresponding states and actions. To achieve this, we minimise the mean squared error between the predicted and target Q-values. This loss, denoted as $\mathcal{L}_Q$, is expressed as:

$$\mathcal{L}_Q = \mathbb{E}_{(s,a,Q) \sim \mathcal{D}} \Big( Q_\theta(s,\, a) - Q \Big)^2 \tag{9}$$

Table 1: Comparison of DTS with the baselines using *Success Rate* and *Collision Rate* on Navigation.

| Solver | Navigation (2-exits) | | Navigation (1-exit) | |
|---|---|---|---|---|
| | Success Rate | Collision Rate | Success Rate | Collision Rate |
| Model-Free Q-Network | 94.5% ($\pm$ 0.2%) | 4.4% | 47.1% ($\pm$ 0.5%) | 50.2% |
| Search with World Model | 93.2% ($\pm$ 0.3%) | 6.7% | 86.9% ($\pm$ 0.3%) | 12.4% |
| TreeQN | 95.4% ($\pm$ 0.2%) | 3.8% | 51.8% ($\pm$ 0.5%) | 39.2% |
| **DTS** | **99.0%** ($\pm$ 0.1%) | **0.7%** | **99.3%** ($\pm$ 0.1%) | **0.2%** |

In the offline-RL setting, however, there's a risk that Q-values for out-of-distribution actions might be overestimated (Kumar et al., 2020). To address this, we incorporate the CQL (Kumar et al., 2020) loss. This additional loss encourages the agent to adhere to actions observed within the training data distribution. This loss, $\mathcal{L}_{\mathcal{D}}$, is defined as:

$$\mathcal{L}_{\mathcal{D}} = \mathbb{E}_{(s,a)\sim\mathcal{D}}\Big( \log \sum_{a'} \exp\Big( Q_\theta(s,\ a') \Big) - Q_\theta(s,\ a) \Big) \tag{10}$$

Based on each method's specifications, we combine the loss functions from equations 7, 8, 9, and 10 during training, as detailed in Section 4.3.

### 4.3 BASELINES AND IMPLEMENTATION DETAILS

To evaluate the efficacy of Differentiable Tree Search , we benchmark it against the following prominent baselines:

1. **Model-free Q-network**: This allows us to assess the significance of integrating the inductive biases into the neural network architecture. This model is trained using the loss defined as, $\mathcal{L}_{Qnet} = \lambda_1\mathcal{L}_Q + \lambda_2\mathcal{L}_{\mathcal{D}}$.

2. **Online Search with Independently Learnt World Model**: In this baseline, the world models and value module are trained independently of each other. During evaluation, we employ a best-first search, akin to DTS, utilising the independently trained world model. Through this baseline, we assess the benefits derived from the joint optimisation of the world model and the search algorithm. For brevity in our discussions, this is termed *"Search with World Model"*. Notably, our evaluations perform 10 trials for each search. To train this model, we compute the Q-value, $Q_\theta$, *without performing the search* and optimise the loss defined as, $\mathcal{L}_{Search} = \lambda_1\mathcal{L}_Q + \lambda_2\mathcal{L}_{\mathcal{D}} + \lambda_3\mathcal{L}_{\mathcal{T}_\theta} + \lambda_4\mathcal{L}_{\mathcal{R}_\theta}$.

3. **TreeQN**: This comparison helps in highlighting the advantages of using a more advanced search algorithm that can execute a deeper search while maintaining similar computational constraints. We adhere to a depth of 2 for evaluations, as stipulated in Farquhar et al. (2018), for both Procgen and navigation domains. Notably, greater depths, such as 3 or more, are infeasible since the resulting computation graph exceeds the memory capacity (roughly 11GB) of a typical consumer-grade GPU. This model is trained using the loss (as discussed in Farquhar et al. (2018)) defined as, $\mathcal{L}_{TreeQN} = \lambda_1\mathcal{L}_Q + \lambda_2\mathcal{L}_{\mathcal{D}} + \lambda_3\mathcal{L}_{\mathcal{R}_\theta}$.

Moreover, we set the maximum search trials for DTS at T=10 and train it using the loss defined as $\mathcal{L}_{DTS} = \lambda_1\mathcal{L}_Q + \lambda_2\mathcal{L}_{\mathcal{D}} + \lambda_3\mathcal{L}_{\mathcal{T}_\theta} + \lambda_4\mathcal{L}_{\mathcal{R}_\theta}$. Every method is trained with same datasets using their respective loss functions. We fine-tune the hyperparameters, $\lambda_1, \lambda_2, \lambda_3$ and $\lambda_4$, using grid search on a log scale. A more comprehensive discussion of the baselines can be found in Section E of the appendix.

### 4.4 RESULTS

**Navigation:** We compare DTS against the baselines using evaluation metrics like success rate and collision rate, where success rate refers to the fraction of test levels completed by the agent, and collision rate refers to the fraction of levels failed due to collision with a wall. We report the evaluation results in the Table 1. We observe that DTS outperforms the baselines on both navigation scenarios. Notably, when agents are trained on data from the 2-exits scenarios but are tested in the

Table 2: Comparison of DTS with the baselines using *Mean Scores* and *Z-Score* on Procgen games

| Games | Model-free Q-network | Search with World Model | | TreeQN | | **DTS** | |
|---|---|---|---|---|---|---|---|
| | Mean Score | Mean Score | Z-score | Mean Score | Z-score | Mean Score | Z-score |
| bigfish | **21.49** | 14.51 | -0.42 | 19.06 | -0.14 | 20.80 | -0.04 |
| bossfight | **8.77** | 6.16 | -0.45 | 8.31 | -0.08 | 8.53 | -0.04 |
| caveflyer | 2.01 | 3.35 | 0.34 | **3.57** | **0.40** | 3.53 | 0.39 |
| chaser | 5.79 | 4.43 | -0.26 | 6.46 | 0.13 | **6.66** | **0.17** |
| climber | 2.81 | **5.34** | **0.56** | 4.34 | 0.34 | 5.20 | 0.53 |
| coinrun | 5.02 | 6.50 | 0.30 | 5.07 | 0.01 | **6.56** | **0.31** |
| dodgeball | 1.10 | 4.66 | 1.87 | 4.79 | 1.94 | **5.52** | **2.33** |
| fruitbot | 14.00 | 10.50 | -0.30 | **15.59** | **0.14** | 13.85 | -0.01 |
| heist | 0.81 | 1.97 | 0.43 | 1.99 | 0.43 | **2.12** | **0.48** |
| jumper | 3.39 | 4.06 | 0.14 | 3.42 | 0.01 | **4.31** | **0.19** |
| leaper | 7.57 | 6.35 | -0.28 | 7.87 | 0.07 | **8.05** | **0.11** |
| maze | 2.00 | 2.63 | 0.16 | 2.18 | 0.05 | **2.51** | **0.13** |
| miner | 1.42 | 1.55 | 0.08 | 1.55 | 0.08 | **1.63** | **0.14** |
| ninja | 5.07 | 5.15 | 0.02 | 5.16 | 0.02 | **5.88** | **0.16** |
| plunder | 12.77 | 9.67 | -0.28 | 12.50 | -0.02 | **13.26** | **0.04** |
| starpilot | 15.53 | 13.57 | -0.15 | **16.98** | **0.11** | 16.91 | 0.11 |
| Mean Z-score | - | | **0.11** | | **0.22** | | **0.31** |

Table 3: Head-to-head comparison of DTS with the baselines using *number of games won* in Procgen.

| Head-to-head Wins | Model-free Q-Network | Search with World Model | TreeQN |
|---|---|---|---|
| Differentiable Tree Search | 13 / 16 | 14 / 16 | 13 / 16 |

1-exit scenario, DTS, with its powerful inductive bias, retains its performance. In stark contrast, the Model-free Q-network and TreeQN experience a substantial performance decline, which underscores their limited generalisation ability in even marginally modified environments. The online search strategy, which employs a separately learnt world model, also registers a minor decrease in the success rate, reinforcing the importance of jointly optimising the world model for enhanced robustness.

**Procgen:** We evaluate the performance of DTS and the baselines on all the games in Procgen suite, comparing their performance in terms of Z-score and head-to-head wins. We also report the mean scores across 1000 episodes obtained by these methods in Table 2. We use model-free Q-network as baseline to compute a normalised score[1], Z-score = $(\mu_\pi - \mu_B)/\sigma_B$, where $\mu_\pi$ and $\mu_B$ represent the mean scores obtained by the agent policy and the baseline policy respectively and $\sigma_B$ represents the standard deviation of the scores obtained by the baseline policy. We observe that DTS reports a superior mean Normalise Score, averaged across the 16 Procgen games. This difference was particularly noticeable in games such as climber, coinrun, jumper, and ninja, which necessitate the planning of long-term action consequences, highlighting the role of the stronger inductive bias employed by DTS. In Procgen, online search strategy with a separately learnt world model lags behind both TreeQN and DTS, highlighting that for complex environments, joint optimisation enables learning a robust world model. In head-to-head comparison listed in Table 3, DTS won in 13 games against the Model-free Q-network, 14 games against online search, and 13 games against TreeQN.

## 4.5 DISCUSSION

**Role of Telescoping Sum Trick and the REINFORCE Term:** In an ablation study, we assess the impact of both the telescoping sum trick and the REINFORCE term on DTS's performance. The results, presented in Table 4, show notable differences. Without the telescoping sum, DTS see a

---

[1]We introduce this normalised score as the baseline normalised score used in prior works Badia et al. (2020); Kaiser et al. (2020); Mittal et al. (2023) is problematic in our experiment, as described in Section E.5

Table 4: Evaluation of the role of *auxiliary losses*, *REINFOCE* term and *telescoping sum* trick on Navigation (1-exit) and Procgen games.

| Solver | Navigation (1-exit) | | Procgen |
|---|---|---|---|
| | Success Rate | Collision Rate | Mean Z-score |
| **DTS** | **99.3%** (± 0.1%) | **0.2%** | **0.31** |
| DTS *without telescoping sum* | 98.5% (± 0.1%) | 1.1% | 0.28 |
| DTS *without REINFORCE term* | 97.7% (± 0.2%) | 0.3% | 0.29 |
| DTS *without auxiliary losses* | 91.1% (± 0.3%) | 7.8% | - |

modest decrease in the success rate for Navigation (1-exit), moving to 98.5% from 99.3%. Similarly, the mean Z-score for Procgen dips to 0.28. The omission of the REINFORCE term also marks a decline, with the Navigation (1-exit) success rate landing at 97.7% and the mean Z-score for Procgen dipping to 0.29.

**Role of Auxiliary Losses:** Our ablation study further explores the contribution of auxiliary losses to DTS's performance. As outlined in Table 4, the absence of these auxiliary losses lead to a more pronounced decline in the performance. Specifically, the success rate in the Navigation (1-exit) task drops significantly to 91.1% compared to the 99.3% achieved when incorporating auxiliary losses. Concurrently, the collision rate rises from a mere 0.2% to 7.8%

**Real-world Training Speed Analysis:** To understand the computational demands of DTS and the baselines in real-world scenarios, we analyse the average time taken per training iteration (both forward and backward passes) for Model-free Q-network, TreeQN, and DTS in the Procgen environment in Table 5. Models were trained on a 2080Ti GPU using PyTorch with a batch size of 256. We observe that DTS provides the ability to conduct a deeper search while still outpacing TreeQN in speed.

Table 5: Comparison of average *runtime* per training iteration on Procgen.

| Solver | Time Taken per Iteration |
|---|---|
| Model-free Q-network | 43ms |
| TreeQN (depth=2) | 357ms |
| TreeQN (depth=3) | Out-of-memory |
| DTS (T=5) | 153ms |
| DTS (T=10) | 294ms |

## 5 CONCLUSION

In this paper, we introduce Differentiable Tree Search (DTS), a novel neural network architecture that conducts a fully differentiable online search in a latent state space. It features four learnable submodules that encode the input state into its latent state and conducts a best-first style search in the latent space using learnt reward, transition, and value functions. Critically, the transition and reward modules are jointly optimised with the search algorithm, resulting in a robust world model that is useful for the online search and a search algorithm that can account for the errors in the world model. Furthermore, we address the potential Q-function discontinuity by employing a stochastic tree expansion policy. We optimise the expected loss function using a REINFORCE-style objective function and proposed a telescoping sum trick to reduce the variance of the gradient for this expected loss. We evaluate DTS against well-known baselines on Procgen games and a navigation task in an offline reinforcement learning setting to assess their sample efficiency and generalisation capabilities. Our results show that DTS outperforms the baselines in both domains.

Nonetheless, the current implementation of DTS has some limitations. Primarily, its strength is currently limited to deterministic decision-making scenarios. To cater to a broader spectrum of decision-making problems, there's a need to revamp the transition model to manage stochastic world scenarios. Additionally, the computation graph in DTS can grow considerably as search trials increase, potentially decelerating training speed and exceeding the memory limits of standard GPUs. In future works, we will focus on refining DTS to facilitate deeper searches efficiently.

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
