# OpenReview forum: "Differentiable Tree Search in Latent State Space"
_ICLR.cc/2024/Conference — ICLR 2024 Conference Withdrawn Submission_

### Official Review · Reviewer_CkbR · 2023-10-26

**Soundness:** 2 fair
**Presentation:** 3 good
**Contribution:** 3 good
**Rating:** 6
**Confidence:** 4

**Summary:**

The paper proposes Differentiable Tree Search (DTS) which improves upon the former TreeQN method. The goal is to combine planning (through tree search in a learned world model) with a learned value function. While TreeQN expands the full tree (with a branching factor of |A|) DTS selectively only expands the most promising nodes, allowing much deeper search depth.
The authors propose an interesting trick to overcome the discontinuity induced by the discrete choice of which node to expand further.
They evaluate the proposed method in an Offline-RL setting on two environments.

**Strengths:**

* The extension to TreeQN of selectively expanding only the most promising nodes makes a lot of sense
* The handling of the discontinuity (using a telescoping sum argument) makes sense and is elegant
* I also find the Offline-RL setting to be a well-chosen benchmark. I think the argument for using inductive bias (such as planning + world model) is much stronger in this setting than in online-RL where data is 'infinite' (in a sense).
* The experimental results seem very strong.

**Weaknesses:**

Please see the "Questions" section where I ask for clarification on some potential weaknesses.

In the experiment section, it would be great to show that the performance of DTS scales with the depth of the search. A core feature of tree search is that computation can be traded off against performance.

**Questions:**

There is two sources of potential biases that I can see, and it would be great to get some clarification on those:
1. While the loss L_Q uses the Q-learning update rule (using the $\max$ operator), the stochastic branch selection is more similar to the (expected) SARSA update rule. I'm wondering if this mis-match is a problem. In particular, is it even possible for the loss L_Q to go to zero: For example, even if we assumed we had learned the correct Q-function, I think L_Q wouldn't be 0?
2. The loss L_D has the known downside that it pushes the action-values of actions not in the dataset to -infty. This is (somewhat) intended, but unlike in typically used learned Q-functions, here the Q-function is represented as a tree search. Hence the question arises of how the DTS Q-function represents the large negative Q-values and to what extends this creates problems?

---

> ### Author Response · Authors · 2023-11-18
> **Rebuttal to the Official Review by Reviewer CkbR**
>
> We greatly appreciate your time and insightful feedback on our manuscript regarding DTS. Your recognition of the sensibility and elegance of our approach, as well as the strength of our results, is encouraging. We address the concerns you raised as follows:
>
> > “In the experiment section, it would be great to show that the performance of DTS scales with the depth of the search ……”
>
> We are grateful for your suggestion to demonstrate how DTS's performance scales with the depth of the search. In response, we conducted additional experiments with varying search trials (5, 10, 20) in the navigation domain. The results clearly indicate an improvement in performance with increased search trials, and correspondingly depth:
>
> | Method | Success Rate (Easy) | Success Rate (Hard) |
> |:---------|:----:|:----:|
> |DTS (trials = 5)   |97.4% | 96.6%|
> |DTS (trials = 10) |99.0% | 99.3% |
> |DTS (trials = 20) | 99.5% | 99.1% |
>
> These findings will be included in the final manuscript, providing concrete evidence of DTS's capability to effectively scale its performance with deeper searches.
>
> > “While the loss L_Q uses the Q-learning update rule (using the operator), the stochastic branch selection is more similar to the (expected) SARSA update rule ……”
>
> The expectation in the equation is applied over the tree construction policy. Once a tree is constructed, the max operator is used to select the best path in the tree starting from the leaves to the root. If a large enough tree is constructed, the optimal path would be selected, resulting in accurate Q-value estimates. For example, in TreeQN’s case if the tree is deep enough then it will select the optimal path by full tree expansion. Now, if the correct Q-function is used in TreeQN, it would give zero loss. Likewise, in DTS, if the search process finds the optimal path, this too would result in zero loss.
>
> > “The loss L_D has the known downside that it pushes the action-values of actions not in the dataset to -infty. This is (somewhat) intended, ……”
>
> Your concern about loss L_D pushing action-values of actions not in the dataset is an important consideration. As you noted, this effect is somewhat intentional. However, we understand the need for clarity on how DTS manages these negative values during tree search. In DTS, the Q-value for each action at the root node is determined by the maximum value trajectory stemming from that action. When loss L_D reduces the action-values of actions not present in the dataset, it consequently lowers the trajectory values associated with these actions. This reduction in trajectory values leads to a dynamic shift in the tree search process. Upon re-computation of Q-values for the same sample, DTS adapts by selecting an alternate trajectory with a higher value, as the previously dominant trajectory's value has been diminished by L_D. During the backup phase of the tree search, the max operator plays a crucial role in propagating values from the leaves to the root. This process inherently filters out the lower Q-values affected by L_D, ensuring they don't adversely influence the overall decision-making at the root node. Furthermore, in the computation of the tree search policy, these lower Q-values translate into minimal probabilities, effectively removing them from consideration in subsequent tree expansions. This mechanism ensures that the impact of negative Q-values is mitigated, maintaining the integrity and effectiveness of the search process.
>
> We deeply value the thoughtful and constructive feedback you have provided. We sincerely hope we have addressed your concerns  and raised your impression of our work. We are committed to enhancing our work based on this insightful feedback and are ready to provide any additional clarifications.

---

> > ### Comment · Reviewer_CkbR · 2023-11-22
> > **Thank you for your reply**
> >
> > Dear authors,
> >
> > Thank you very much for your reply and the additional experiments, that's great results to see. For these experiments, did you vary the number of trials only during inference, or also during training?

---

> > > ### Author Response · Authors · 2023-11-23
> > > **Further clarifications**
> > >
> > > Dear Reviewer CkbR,
> > >
> > > We are delighted that you found the additional results interesting. To answer your query, we maintained a consistent number of trials during both the training and inference phases. Specifically, when we trained the model on T search trials, then we tested it on the same number of trials. This approach ensures that the input distribution for the DTS submodules remains consistent.
> > >
> > > Moreover, we conducted a supplementary study to evaluate the robustness of the learned world models when subjected to deeper searches than those used during the training phase. For instance, we trained the model on 10 trials but increased the trials to 20 and 50 during inference. The results demonstrate improvement in performance with an increased search budget, highlighting the robustness of the world models learnt through end-to-end learning. The following table presents these findings:
> > >
> > > | Method | Success Rate (Easy) | Success Rate (Hard) |
> > > |:---------|:----:|:----:|
> > > |DTS (trials = 10) |99.0% | 99.3% |
> > > |DTS (trials = 20) | 99.3% | 99.4% |
> > > |DTS (trials = 50)   |99.7% | 99.6%|
> > >
> > > Additionally, we performed a similar analysis on world models learned via non end-to-end (or independently learned) model-based RL techniques. We observed that these models do not demonstrate sufficient robustness to support deeper searches, with their performance deteriorating due to compounding errors. The results of this analysis are as follows:
> > >
> > > | Method | Success Rate (Easy) | Success Rate (Hard) |
> > > |:---------|:----:|:----:|
> > > |Search with World Model (trials = 10) |93.2% | 86.9% |
> > > |Search with World Model (trials = 20) | 91.1% | 84.6% |
> > > |Search with World Model (trials = 50)   |89.5% | 80.4%|
> > >
> > > These findings significantly contribute to the strength of our paper, and we are grateful to you for prompting this aspect of the discussion. We will include these important results in the updated manuscript.

---

### Official Review · Reviewer_ZXWN · 2023-10-30

**Soundness:** 3 good
**Presentation:** 3 good
**Contribution:** 3 good
**Rating:** 6
**Confidence:** 4

**Summary:**

The paper introduces a novel architecture to perform differentiable tree search (DTS) in the latent space of a world model. This world model, in conjunction with the search algorithm, is optimized to produce a robust outcome and minimize model inaccuracies. The authors also tackle potential discontinuities in the Q-function by using a stochastic tree expansion policy and reduce the gradient variance by computing the REINFORCE objective using a telescoping sum. When evaluated on Procgen games and a grid navigation task, DTS surpassed both model-free and model-based baselines in performance.

**Strengths:**

1. The methodological contributions of the paper seem novel and interesting. The paper adopts the sophisticated search machinery used in the search literature within a TreeQN like network. The paper further proposes several solutions to mitigate the numerical issues that arise as a result.
2. The paper is also reasonably clearly written overall. Although, I believe the related work section could be improved to better distinguish the related work from the paper’s work.
3. The results show a convincing improvement over the chosen baselines. The baselines chosen do a good job of showing the impact of the specific improvements suggested by the paper. Furthermore, the ablation experiments provide a good analysis of the impact of the Reinforce term, telescoping sum and the auxiliary losses used.

**Weaknesses:**

1. The paper shows experiments only with optimal demonstrations in the offline dataset. However, given the broader appeal for offline RL, I would have appreciated experiments with sub-optimal demonstrations as well. I would expect the stochastic tree search to have higher variance during training hence making it harder to train. However, I would also expect the tree search inductive bias to be especially useful in that setting. Thus, it would be interesting to see the trade-offs involved and some related analysis.
2. As mentioned in the previous section, I would appreciate it if the related work section is rephrased slightly to better distinguish the related work from the current work.

**Questions:**

Included in the above sections

The authors report the training time per iteration. I would appreciate it if the authors report the full training time and the Inference time separately. I would expect the additional stochasticity to slow the overall training time. Eitherway it would help to show them separately and include them in the discussion

---

> ### Author Response · Authors · 2023-11-18
> **Rebuttal to the Official Review by Reviewer ZXWN**
>
> Firstly, we express our gratitude for your thorough review and insightful comments on our paper. Your recognition of the novelty and clarity of our work is highly appreciated, and we have taken your feedback as an opportunity to further refine our manuscript. We address your concerns below:
>
> > “The paper shows experiments only with optimal demonstrations in the offline dataset. However, given the broader appeal for offline RL, ……”
>
> We want to emphasise that DTS is not constrained to optimal data; it is fully compatible with data from sub-optimal policies as well. However, we think that the effects of the search inductive bias are best investigated in the presence of optimal demonstrations without the interference of noise. We agree that it would be interesting to further investigate the effects of noisy demonstrations on DTS.
>
> > “As mentioned in the previous section, I would appreciate it if the related work section is rephrased slightly to better distinguish the related work from the current work.”
>
> We appreciate your suggestion to improve the clarity of the related work section. In the final manuscript, we will revise this section to more clearly distinguish our contributions from existing literature.
>
> > “The authors report the training time per iteration. I would appreciate it if the authors report the full training time ……”
>
> Thank you for suggesting the inclusion of full training and inference time data. We agree that this information is crucial for a complete understanding of our method's practical implications.
> Here are the average training times for a single run on Procgen:
>
> |          Method      |             Time        |
> |:----------------|:-------------------|
> | DTS (trials=10)  | 17 hours 30 minutes |
> | TreeQN (depth=2) | 23 hours 45 minutes |
> | Model-free       | 4 hours             |
>
> In the final manuscript, we will include detailed time metrics for DTS, alongside a comparative analysis with other methods. This will offer readers a comprehensive view of the computational efficiency and real-world applicability of our approach.
>
> We sincerely hope we have addressed your concerns with the above discussion and planned modifications to the manuscript and raised your impression of our work. We are dedicated to improving our work based on your valuable feedback and are available for any further clarifications.

---

> > ### Comment · Reviewer_ZXWN · 2023-11-22
> > **Official comment by Reviewer ZXWN**
> >
> > Thanks for the clarifications. While the authors claim "we think that the effects of the search inductive bias are best investigated in the presence of optimal demonstrations without the interference of noise. We agree that it would be interesting to further investigate the effects of noisy demonstrations on DTS.", I would have liked to actually see experimental results and some analysis along those lines to say the paper was a clear accept. Thus, I'm keeping my score unchanged.

---

> > > ### Author Response · Authors · 2023-11-23
> > > **Further Clarifications**
> > >
> > > Dear Reviewer ZXWN,
> > >
> > > We would like to draw your attention to the demonstration policy used to collect data for the Procgen environments. This demonstration policy is, in fact, a sub-optimal policy, scoring on average lower than the original PPG results (as illustrated in the table below). Therefore, the Procgen environment does include noisy demonstrations from a sub-optimal policy. Despite this, DTS still demonstrates gains over the baselines, even within the context of sub-optimal policy demonstrations.
> > >
> > > | Game | Demonstration Policy | PPG (Github Repo) |
> > > |:---------|:----:|:----:|
> > > | Bigfish   | 26.98 | 34.38 |
> > > | Bossfight | 11.13 | 12.25 |
> > > | Caveflyer| 9.66 | 11.55 |
> > > | Chaser| 8.11 | 10.07 |
> > > | Climber| 9.95 | 11.05 |
> > > | Coinrun| 8.40 | 9.55 |
> > > | Dodgeball| 8.88 | 13.57 |
> > > | Fruitbot | 23.52 | 25.81 |
> > > | Heist | 3.49 | 6.90 |
> > > | Jumper | 6.54 | 7.50 |
> > > | Leaper | 8.92 | 9.62 |
> > > | Maze | 7.00 | 9.25 |
> > > | Miner | 10.84 | 13.11 |
> > > | Ninja | 9.41 | 9.77 |
> > > | Plunder | 17.67 | 17.65 |
> > > | Starpilot | 22.33 | 22.87 |
> > >
> > > We want to thank you for bringing this point in the discussion and we will ensure to include a note regarding this point in the updated version of our manuscript.
> > >
> > > We are grateful for your insights and hope that our work, in its current form, has provided valuable contributions to the field and offered new learnings. Your confidence in our work is deeply appreciated, and we look forward to enhancing our manuscript as per your suggestions.
> > >
> > > Thank you very much!

---

### Official Review · Reviewer_NKWY · 2023-10-30

**Soundness:** 3 good
**Presentation:** 2 fair
**Contribution:** 3 good
**Rating:** 6
**Confidence:** 3

**Summary:**

The authors propose a novel neural network architecture that incorporates the structural bias of a best-first search algorithm into the network design. The authors evaluate the proposed architecture in an offline-RL setting with a limited training data scenario on Procgen games and grid navigation task, and demonstrate that DTS outperforms popular model-free and model-based baselines. The computer experiments include ablation studies and comparisons with baselines.

**Strengths:**

- The paper presents a comprehensive experimental evaluation of the proposed architecture in an offline-RL setting with a limited training data scenario on Procgen games and grid navigation task, and demonstrate that DTS outperforms popular model-free and model-based baselines. The computer experiments include ablation studies and comparisons with baselines.

**Weaknesses:**

- The text lacks an illustration of the DTS architecture. An image or scheme of the DTS architecture would be very helpful.
- It seems that the authors are assuming familiarity of the reader with other algorithms (A* search, TreeQN, etc). This assumption hurts the self-containedness of the paper. The authors should provide a brief description and/or illustration of the algorithms used in the paper.

**Questions:**

- The authors say "Differentiable Tree Search (DTS) is a neural network architecture that incorporates the structural bias
of a best-ﬁrst search algorithm into the network design". How is this achieved? What is the structural bias of a best-first search algorithm? In my understanding, this is the main contribution of the paper, but it is not clear how this is achieved. An explanation of how a best-first search algorithm works and what are the parallels with the DTS architecture would be extremely helpful. Ideally, an image or scheme of the DTS architecture would be very helpful.
- (3 Differentiable Tree Search) The authors provide a good description of the DTS architecture. However, it is not clear how the DTS architecture is trained.
An image or illustation of the DTS architecture would be extremely helpful to understand the model. The image could add the different submodules of the model and how they are connected.
- (4.1 Test Domains) The authors evaluate DTS in discrete action spaces. Can the authors comment on the applicability of DTS to continuous action spaces?

---

> ### Author Response · Authors · 2023-11-20
> **Rebuttal to the Official Review by Reviewer NKWY**
>
> We thank you, NKWY, for dedicating your time and effort in reviewing our work on DTS. We are pleased to know that you find our experimental evaluation comprehensive and the DTS architecture novel. We address the concerns you raised as follows:
>
> > “The text lacks an illustration of the DTS architecture ……”
>
> Acknowledging your excellent suggestion, we have prepared an illustration of the DTS architecture, which will be accessible via an anonymous link for your reference. We plan to incorporate this illustration in the final manuscript to offer a clear visualisation of the DTS architecture and its operational mechanics. We believe this will significantly aid in understanding the structure and functionality of DTS for the future readers.
>
> > “It seems that the authors are assuming familiarity of the reader with other algorithms ……”
>
> We recognise your concerns about the self-containedness of our paper and our assumption of reader familiarity with certain algorithms. To address this, we will enhance the appendix with a detailed background section, including descriptions and illustrations of the algorithms used. Additionally, improvements will be made to the related works section in the main paper to provide better clarity for new readers, ensuring a comprehensive understanding of the context and relevance of our work.
>
> > “The authors say 'Differentiable Tree Search (DTS) is a neural network architecture that incorporates the structural bias of a best-first search algorithm into the network design.' How is this achieved? ……”
>
> We acknowledge your inquiry regarding the integration of a best-first search approach in constructing the DTS computation graph and executing a differentiable search. We agree that visualising this process would greatly aid in understanding how DTS works. Therefore, we have prepared an illustrative depiction of the computation graph's construction and will include it in the final manuscript. We will share it in the anonymous link for your reference. To provide a preliminary understanding, the computation graph in DTS can be thought of as an advanced version of typical neural network architectures. Typically, neural networks process inputs through a series of Conv or Linear layers sequentially, leading to the output, such as Q-values in our case. These outputs are then assessed via a loss function, and the network is optimised through backpropagation across each layer.
>
> DTS, however, diverges from this linear sequence. It employs a best-first search strategy to assemble trainable modules, crafting a computation graph uniquely tailored for outputting Q-values. This structure allows for a more strategic flow of data and gradients than the standard sequential models. During the backpropagation phase, DTS fine-tunes each module within this framework, enhancing the accuracy of Q-value predictions. Simultaneously, it ensures that each module contributes effectively to the best-first search algorithm, optimising the entire architecture for both accuracy and search efficiency
>
> > “The authors provide a good description of the DTS architecture. However, it is not clear how the DTS architecture is trained ……”
>
> We appreciate your suggestion to include an illustration detailing the training process and construction of the DTS architecture. We will add an explanatory figure to the final manuscript.
>
> > “The authors evaluate DTS in discrete action spaces. Can the authors comment on the applicability of DTS to continuous action spaces?”
>
> The current implementation of DTS is specifically tailored for discrete action spaces, utilising a tree-based search algorithm. Adapting DTS for continuous action spaces would require modifications to its algorithm to construct continuous policy distributions and sample actions accordingly. We acknowledge this as an area for future work and plan to explore these adaptations in subsequent research.
>
> We sincerely appreciate the constructive critique you have provided and hope that our responses and the planned modifications to the manuscript adequately address your insightful feedback and increases your impression and confidence in our work. We are dedicated to refining our work based on your valuable comments and remain available for any further clarifications.

---

> ### Author Response · Authors · 2023-11-23
> **DTS Illustrations**
>
> Dear Reviewer NKWY,
>
> We are providing the anonymous links for the illustrations that further clarifies various aspects of our DTS architecture:
>
> - Illustrating the input/output to submodules: https://ibb.co/p3qTnpQ
> - Illustrating how DTS utilises best-first search to compute Q-values:  1.  https://ibb.co/rxqzKd6  2.  https://ibb.co/bWkq5ds. 3.  https://ibb.co/wNcLWNs
> - Illustrating the construction of the corresponding computation graph: https://ibb.co/k5QZztk
>
> We will add these illustrations to the updated manuscript.

---

### Official Review · Reviewer_SaqX · 2023-11-01

**Soundness:** 3 good
**Presentation:** 2 fair
**Contribution:** 2 fair
**Rating:** 5
**Confidence:** 4

**Summary:**

This paper presents a method for incorporating tree search within a policy optimization method. At a high level this work builds upon TreeQN and incorporates a best-first search algorithm for expanding the tree as opposed to the breadth-first approach taken by TreeQN. Whereas TreeQN expands all nodes up to a fixed depth, and thus has exponential complexity based on the branching factor, the primary contribution of this work is to selectively expand nodes based on their expected value which is not subject to the same blow-up in complexity and allows the algorithm to evaluate deeper. To do so in a way that isn't subject to discontinuities the authors expand nodes by sampling proportional to the softmax of the expected value of the entire path.

The authors also present a number of other improvements including a variance reduction technique based on a telescoping sum as noted by Guez et al. which takes advantage of the fact that by expanding the loss include repeated terms that add variance but are zero in expectation.

Similarly the authors apply their approach to problems in Batch RL using examples from an optimal policy, and hence they also introduce a term from CQL to address Q-function overestimation for infrequently visited states.

Their results show that these techniques improve over the various baselines on a number of procgen tasks.

**Strengths:**

The idea is interesting and provides a good way to address one of the shortcomings of TreeQN, namely the complexity of dealing with deep trees. Similarly the application to procgen is good, and departs from the more standard atari tasks, and the experiments do show improvements over the baselines. The combination of the additional loss components is also novel (as far as I am aware) albeit rather straightforward.

**Weaknesses:**

I have a few criticisms of this work:

(1) The primary contribution (e.g. the mechanism by which nodes are expanded in contrast to TreeQN) while novel, is marginally novel. This is addressing a real problem, but it's not clear by looking at the results that it gives a significant gain, although it clearly does provide gains. The additional minor contributions (e.g. the telescoping variance reduction technique and the CQL loss) are similarly minor. The first follows from other standard procedures for reducing variance for REINFORCE-style algorithms and the second is necessary but is one of a standard family of approaches which addresses the batch-RL setting (see my next point).

(2) It is unclear why the authors are specifically addressing the batch RL setting. It would in my opinion be much more interesting to apply this to the online or at least growing batch setting. When learning the latent transition models and using them to inform how the tree is expanded the use of data from an optimal policy is particularly strong, and potentially more simple techniques based on behavioral cloning could be applied. This adds a new dimension that I'm not sure is necessary to compare against the original technique (and requires more info on how the data is generated, etc.).

(3) Finally while the presentation of the work is reasonable there are a number of points where the the fundamental algorithm being discussed is not quite clear. In particular the pseudocode is only included in the appendix, and even the definition of the loss being optimized (e.g. the TD-error) is only defined in the appendix. Similarly the precise definition of what a node is is unclear, and things like the latent state are represented as h_t or h_n depending on whether this is depth or node. This can be inferred, but this section should be given another pass to make it much more explicit about what the algorithm is doing.

**Questions:**

See the above. Particularly the first two points, whereas the third is something the authors should definitely address, but I'm fairly confident that they would be able to do so.

Overall I do like the idea, but fundamentally I feel the authors should address the question of why the batch RL setting is the right setting to be making this comparison (e.g. with TreeQN).

---

> ### Author Response · Authors · 2023-11-18
> **Rebuttal to the Official Review by Reviewer SaqX**
>
> We thank you, SaqX, for your insightful and constructive feedback on our manuscript. We appreciate your recognition of the novelty and practical application of our approach in addressing the complexity issue in TreeQN and its application to procgen tasks. Below, we address your concerns to clarify and improve our work.
>
> > “The primary contribution (e.g. the mechanism by which nodes are expanded in contrast to TreeQN) while novel, is marginally novel .……”
>
> We indeed build upon the excellent technical contributions of previous works in DTS. However, we would like to point out the conceptual novelty and the practical novelty of our contributions. Conceptually, we demonstrate how a deeper search can be learned with moderate cost using the DTS architecture. Practically, we show that the gains of deeper search can be substantial in some domains, particularly in the Navigation domain, as shown in Table 1, and that consistent gains can be achieved in other domains, notably the Procgen domain.
>
> > “It is unclear why the authors are specifically addressing the batch RL setting. It would in my opinion be much more interesting to apply this to the online or at least growing batch setting ……”
>
> > “Overall I do like the idea, but fundamentally I feel the authors should address the question of why the batch RL setting is the right setting to be making this comparison (e.g. with TreeQN).”
>
> In our opinion, the gains from the search inductive bias are most effectively explored in an offline RL setting, particularly with optimal demonstrations without the interference of noise. In the online RL setting, there would be a need to disentangle the effects of exploration. However, we acknowledge that it would be interesting to investigate these effects in an online RL context, where inductive bias interplays with exploration and potentially noisy demonstrations.
>
> Regarding the training data, details about the data collection process are currently outlined in Section E.3 of the appendix. In response to your feedback, we plan to enhance this section in the final manuscript for a more comprehensive and clear presentation of our training setup.
>
> > “Finally while the presentation of the work is reasonable there are a number of points where the fundamental algorithm being discussed is not quite clear ……”
>
> We are grateful for your suggestions on improving the presentation of our paper. To address the clarity issues, we will revise the final manuscript to include a more detailed explanation of key terms such as loss functions, nodes, and latent states. We will make efforts to ensure that our algorithm's description is clear and comprehensible, even to readers less familiar with the specific technicalities of our field.
>
> We deeply value the constructive critique that you have provided and sincerely hope that we have addressed your insightful feedback and raised your impression of our work. We will be happy to clarify further if needed.